# Photorelaxation Pathways of 4-(*N*,*N*-Dimethylamino)-4′-nitrostilbene Upon S_1_ Excitation Revealed by Conical Intersection and Intersystem Crossing Networks

**DOI:** 10.3390/molecules25092230

**Published:** 2020-05-09

**Authors:** Ziyue He, Ruidi Xue, Yibo Lei, Le Yu, Chaoyuan Zhu

**Affiliations:** 1Key Laboratory of Synthetic and Natural Functional Molecule of the Ministry of Education, College of Chemistry & Materials Science, Shaanxi key Laboratory of Physico-Inorganic Chemistry, Northwest University, Xi’an, Shaanxi 710127, China; heziyue@stumail.nwu.edu.cn (Z.H.); xueruidi@stumail.nwu.edu.cn (R.X.); 2Institute of Molecular Science and Department of Applied Chemistry, National Chiao Tung University, Hsinchu 30010, Taiwan; 3Center for Emergent Functional Matter Science, National Chiao Tung University, Hsinchu 30010, Taiwan

**Keywords:** photorelaxation, conical intersection, intersystem crossing, MS-NEVPT2

## Abstract

Multi-state *n*-electron valence state second order perturbation theory (MS-NEVPT2) was utilized to reveal the photorelaxation pathways of 4-(*N,N*-dimethylamino)-4′-nitrostilbene (DANS) upon S_1_ excitation. Within the interwoven networks of five S_1_/S_0_ and three T_2_/T_1_ conical intersections (CIs), and three S_1_/T_2_, one S_1_/T_1_ and one S_0_/T_1_ intersystem crossings (ISCs), those competing nonadiabatic decay pathways play different roles in *trans*-to-*cis* and *cis*-to-*trans* processes, respectively. After being excited to the Franck–Condon (FC) region of the S_1_ state, *trans*-S_1_-FC firstly encounters an ultrafast conversion to quinoid form. Subsequently, the relaxation mainly proceeds along the triplet pathway, *trans*-S_1_-FC → ISC-S_1_/T_2_-*trans* → CI-T_2_/T_1_-*trans* → ISC-S_0_/T_1_-*twist* → *trans*- or *cis*-S_0_. The singlet relaxation pathway mediated by CI-S_1_/S_0_-*twist-c* is hindered by the prominent energy barrier on S_1_ surface and by the reason that CI-S_1_/S_0_-*trans* and CI-S_1_/S_0_-*twist-t* are both not energetically accessible upon S_1_ excitation. On the other hand, the *cis*-S_1_-FC lies at the top of steeply decreasing potential energy surfaces (PESs) towards the CI-S_1_/S_0_-*twist-c* and CI-S_1_/S_0_-DHP regions; therefore, the initial twisting directions of DN and DAP moieties determine the branching ratio between *α*_C=C_ twisting (*cis*-S_1_-FC → CI-S_1_/S_0_-*twist-c* → *trans*- or *cis*-S_0_) and DHP formation relaxation pathways (*cis*-S_1_-FC → CI-S_1_/S_0_-DHP → DHP-S_0_) on the S_1_ surface. Moreover, the DHP formation could also take place via the triplet relaxation pathway, *cis*-S_1_-FC → ISC-S_1_/T_1_-*cis* → DHP-T_1_ → DHP-S_0_, however, which may be hindered by insufficient spin-orbit coupling (SOC) strength. The other triplet pathways for *cis*-S_1_-FC mediated by ISC-S_1_/T_2_-*cis* are negligible due to the energy or geometry incompatibility of possible consecutive stepwise S_1_ → T_2_ → T_1_ or S_1_ → T_2_ → S_1_ processes. The present study reveals photoisomerization dynamic pathways via conical intersection and intersystem crossing networks and provides nice physical insight into experimental investigation of DANS.

## 1. Introduction

Molecules with π-conjugated moieties are widely used as photochromic probes and light-driven molecular motors for their peculiar photo-induced isomerization towards the ethylenic bridge [1,2,3,4,5]. As a representative model, stilbene serves as the parent moiety in series of photoswitches [6,7,8,9,10,11]. Upon excitation to the S_1_ state, both *trans*- and *cis*-stilbene evolve along the C=C torsion coordinate and decay via twisted S_1_/S_0_ conical intersections (CIs) [12], while for substituted stilbene, the triplet route mediated by intersystem crossing (ISC) may open [13,14], and the formation of stable intramolecular charge transfer (ICT) states evidently affect the fluorescence efficiency [13,14,15,16,17,18,19]. Therefore, the optical properties of stilbenes can be controlled by introducing suitable substitution groups [8]. For example, the nitro, cyano and halogen substitution on the phenyl ring promotes the triplet pathway [20,21,22]; however, the amino group substitution raises the C=C torsion barrier and evidently slow down the isomerization process [23,24].

The 4-(*N*,*N*-dimethylamino)-4′-nitrostilbene (DANS) is a typical example of the so called “push–pull” chromophore, which has electron donor (D) and acceptor (A) groups simultaneously. In recent decades, the DANS has attracted great interests for its applications in nonlinear optics (NLO) [25,26] as second-harmonic generators [27] and waveguide electro-optical modulators [28,29] and in organic light-emitting diodes (OLEDs) as emitting color tuners [30]. The ground state *trans*-DANS possesses a neutral electronic structure with the delocalized π electrons covering the entire molecule. Upon photoexcitation, the electronic structure converts to highly polarized zewitterionic form yielding the planar ICT (PICT) state, which is also known as the locally excited (LE) state [13,14,15,31,32,33,34]. Subsequently, twisting towards central double bond or D/A groups populate the non-fluorescent rotamer or fluorescent twisted ICT (TICT) state, respectively [17]. As a fact of the mixing and interconversion of the conjugated and polarized electronic structures, the complex relaxation dynamics of excited-state DANS are extraordinarily sensitive to the surrounding polarities [35,36,37,38]. The fluorescence quantum yield of DANS increases with solvent changes from non-polar to slightly polar, but then decreases to nearly zero with further increases of solvent polarity. On the other hand, the Stokes shift increases synchronously with the solvent polarity, whereas the photoisomerization quantum yield (*Φ*_iso_) decreases monotonously. These phenomena were attributed to the interplay of various relaxation pathways [13,14,15,31,32,33,34,35,36,37,38,39,40,41,42,43,44,45]. As proposed on spectroscopy studies in nonpolar solvents [9,10,11], the spin-orbit coupling (SOC) between singlet and triplet states for the transoid intermediate (1*t**) is strong enough to populate the transoid triplet conformer (3*t**) via ISC. Along the efficient triplet-state C=C bond twisting route, another ISC from the triplet (3*P**) to singlet (1*P*) states takes place at the perpendicular configuration and yields to either *trans* or *cis* ground-state isomer with equal probabilities. In slightly polar solvents, fluorescence quantum yields of DANS increase to ~0.5, which is attributed to the formation and radiation decay of the second transoid S_1_ conformer *A** via torsion of dimethylamilino and/or nitrophenyl moieties [14,41]. Meanwhile, the S_1_ state C=C bond twisting barrier between planar and perpendicular configurations reduces the isomerization yield. In polar solvents, the non-radiative *A** becomes more stable for enhanced interactions with solvents; hence both *trans ↔ cis* isomerization and fluorescence are eliminated [27,28,29]. The *A** is proposed to be a TICT state; however, the leading twisting coordinate has been debated for decades. In DANS, the nitro, nitrophenyl, dimethylamino and dimethylaminophenyl groups could twist towards linking bond to perpendicular configuration. Based on the comparative study on DANS and its structural analogues, the formation of the TICT state in polar solvents was attributed to the nitro group torsion [35,36], and the relaxation time constant was suggested to be several picoseconds on time-resolved spectroscopy studies [32,33]. In contrast, studies on torsional hindered aminostilbenes suggest that the twisting of phenylene-amino C–N bond yields to the TICT state [43]. Recently, the combined transient absorption spectroscopy and density functional theory (DFT) calculation studies [37,38] agreed that the nitrophenyl torsion is the dominate relaxation process, as disclosed in previous semiempirical calculation [46].

Comparing with sustained experimental interests on DANS, the theoretical studies are limited [36,37,38,46,47,48,49,50,51,52,53,54,55,56]. Early theoretical studies with semiempirical methods [36,47,48,49,50], i.e., AM1, PM3, MNDO and INDO, have suggested the planar ground state *trans*-DANS. Coupled with the configuration interaction technique, those semiempirical methods were utilized to predict the vertical excitation energies (VEEs), dipole moments and oscillator strengths [41,42,43]. In the semiempirical SAM1 study by Farztdinov and Ernsting [46], the solvent polarity dependence of the twisted intermediates towards the five moieties (C=C, nitro, nitrophenyl, dimethylamino and dimethylaminophenyl) was presented. The variation of energy sequences affect the fluorescence quantum yields, ISC, and *trans ↔ cis* isomerization of DANS after excitation to S_1_. Recently, the DFT and time-dependent DFT (TDDFT) calculations were performed for DANS to explore the experimental results of vibrational spectroscopy [52], two photon absorption [53], protonation effects [54] and reduction reaction [55]. Based on the potential energy curves (PECs) along different rotation coordinates, the mechanisms for the formation of TICT states were analyzed [37,38] and the evidences for possible CI and ISC regions were also revealed [37].

Although great efforts have been devoted to uncovering the nature of this unique push–pull chromophore, the discrepancies within the proposed mechanisms clearly indicated that deeper understanding on this molecule is desired, especially from the theoretical point of view. To establish the networks for nonadiabatic decay, the multi-reference computational methods should be employed [57,58], and which are capable of interpreting the interplay of electronic configuration and conformation conversion accompanied with the relaxation process. In this work, we have optimized the stationary geometries, CIs and ISCs within the S_0_, S_1_, T_1_ and T_2_ states of DANS and presented the possible photorelaxation pathways by multi-state *n*-electron valence state second order perturbation theory (MS-NEVPT2) method. The rest of the paper is organized as follows. Section 2 briefly describes the computational methods. Section 3 displays all the optimized geometries that participate in the relaxation dynamics and the analyses on the possible relaxation pathways. Concluding remarks and future prospects are given in Section 4.

## 2. Theoretical Methods and Computational Details

In this work, the lowest three singlet and three triplet states of DANS were considered in the state-average procedure of complete active space self-consistent field (CASSCF) [59,60] with equal weights (SA6-CASSCF). The minima were optimized by the BDF program [61], and the transition states (TS), CIs (S_1_/S_0_ and T_2_/T_1_) and ISCs (S_1_/T_2_, S_1_/T_1_ and T_1_/S_0_) were optimized by the default procedure within the MOLPRO 2009.1 program [62]. The CASSCF method only considers static correlation energy without dynamic correlation correction and then evidently overestimates the vertical excitation energies, as seen in the results for *trans*-form DANS shown in Appendix A. It cannot properly revel the energy evolutions along possible relaxation processes. Therefore, for the SA6-CASSCF optimized geometries, the dynamic correction energies were calculated by the MS-NEVPT2 method implemented in the Xi’an-CI program [63,64] by employing molecular orbital integrals from the BDF program [61]. The 6-31G* basis set [65,66] was used for all atoms in the calculations. The SOC matrices were computed by using the state interaction approach with the Breit–Pauli Hamiltonian (H_BP_) in MOLPRO 2009.1 program [62]. To qualitatively analyze the photorelaxation pathways within the CI and ISC networks, we calculated the one-dimensional potential energy curves for the six coupled electronic states by linear interpolation of internal coordinates (LIIC) at the MS-NEVPT2 level (CASSCF results are given in the Appendix A).

The accuracy of CASSCF calculation relies on the active space, and the CASSCF orbitals are the basis for MS-NEVPT2 calculation; therefore, the selection of active space is the most important procedure before performing the calculation study. The CASSCF calculation with full valence active space that consists of all the valence orbitals and electrons for DANS is computationally unaffordable. Therefore, the orbital properties and performance on vertical excitation energy prediction of some common reduced active spaces, such as CAS (10,8), CAS (10,10), CAS (12,12) and CAS (18,12) were investigated, and finally the CAS (18,12) was chosen with consideration of both the computation efficiency and accuracy. Based on the active molecular orbitals of those active spaces for *trans*-S_0_ and *cis*-S_0_ presented in Appendix A
Appendix A, it is evident that the corresponding orbitals of CAS (18,12) were in agreement with CAS (10,8) but different from CAS (10,10) and CAS (12,12). The latter orbitals were more balanced in the treatment of π and π* orbitals; however, CAS (18,12) for *trans*-S_0_ had four σ orbitals and more electrons distributed on the electron withdrawing group side than those on the electron donating group side. On the other hand, CAS (18,12) for *cis*-S_0_ had only one σ orbital, and the remaining π and π* orbitals were delocalized with electron distributions on both sides. It tended to attribute the different electron distributions of *trans*-S_0_ and *cis*-S_0_ to the molecular geometry. The planar geometry of *trans*-S_0_ led to strong electronic transfer, as shown in Appendix A
Appendix A for active orbitals of CAS (18,12) and CAS (10,8), while the nonplanar geometry of *cis*-S_0_ hindered this transfer and maintained a more delocalized distribution. The vertical excitation energies from *trans*-S_0_ to *S*_1_ calculated by MS-NEVPT2 with reference functions from CAS (10,10) and CAS (12,12) were 4.16 eV and 5.01 eV, respectively, which were remarkably larger than 3.62 eV from CAS (10,8) and 3.53 eV from CAS (18,12) and then 2.97 eV from experiments in cyclohexane [14]. These evidently different S_0_ → S_1_ VEEs by respective active space can be explained by the electronic configuration of S_1_ state. For both CAS(18,12) and CAS(10,8), the main configuration of S_1_ corresponded to the HOMO → LUMO excitation with the weight of ~0.70, however, which reduces to only 0.51 in CAS(10,10) and gives rise to high excitation energy of 4.16 eV. Conversely, for CAS(12,12), the main configurations of S_1_ are HOMO → LUMO+2 (0.24), HOMO-3 → LUMO+1 (0.18) and HOMO-2 → LUMO+2 (0.12), respectively. The main configurations of S_1_ state of *trans*-S_0_ do not contain HOMO → LUMO excitation, while this excitation is one of the main configurations of S_2_ state and the energy gap between S_1_ and S_2_ is only 0.17 eV so that they are quasi-degenerate. The strong coupling between these two states may lead to the energy order change of them. Even though the high-lying S_2_ is assigned as S_1_, the high vertical excitation energy of 5.18 eV attributes to the small weight of HOMO → LUMO excitation (0.20). Since the initial excitation energy plays an important role in the photochemical reaction, the energy of CAS (18,12) was the closest one to experimental observation so that this active space was feasible for computations in this work.

In order to identify the availability of basis set 6-31G*, we have also performed comparative calculations with larger basis sets of 6-311G* and 6-311+G* for S_0_ → S_1_ VEEs at *trans*-S_0_ by MS-NEVPT2 with reference wavefunctions from SA6-CASSCF(18,12). The S_0_ → S_1_ VEEs by 6-31G*, 6-311G* and 6-311+G* are 3.53 eV, 3.37 eV and 3.14 eV, respectively, which are gradually approaching the experimental value of 2.97 eV with enlarging basis sets. Although the small basis set 6-31G* leads to 0.39 eV overestimation from that of 6-311+G*, which agrees with the convergence trend to experimental value by extending basis set. With the consideration of computations time cost by employing those two larger basis sets, 6-31G* is available basis set in this work.

## 3. Results and Discussion

The DANS is composed of five basic moieties, namely, olefinic double bond (DB), *N*,*N*-dimethylamino (DMA), *N*,*N*-dimethylaminophenyl (DAP), nitro (NT) and nitrophenyl (NP). The definition of important internal coordinates correlating with the photorelaxation processes and atomic numbering are given in Scheme 1. The SA6-CASSCF(18,12)/6-31G* optimized internal coordinates for the minimum energy geometries, such as *trans*-, *cis*-, *twist*-DANS, dihydrophenanthrene (DHP) states and ground state TS are listed in Table 1 and those for the CIs and ISCs are given in Table 2. The MS-NEVPT2 calculated potential energies on SA6-CASSCF optimized geometries for six interested states with respect to *trans*-S_0_ are presented in the Table 3 and Table 4 for stable geometries and crossings, respectively. The relative potential energies calculated by SA6-CASSCF and Cartesian coordinates for all the optimized geometries were given in Appendix A.

### 3.1. Minimum Geometries in trans-, cis- and twist-DANS and DHP Form

By employing the CASSCF (18,12) calculation, the *trans*-form DANS minima were optimized in S_0_, S_1_, T_1_ and T_2_ states; however, only in S_0_ and T_2_ states could the *cis*-form minima be observed, and alternatively, the *twist*-form minima were obtained in S_1_ and T_1_ states. Furthermore, the ring-closing products of *cis*-DANS, 4a,4b-DHP conformers were optimized in all involved states. All the *trans*-minima were planar due to the conjugated π orbitals that covering the entire molecule. In agreement with previous semiempirical and DFT calculations [37,38,46], the C7–C8 was a double bond with length of 1.351 Å in optimized *trans*-S_0_; the NP and DAP moieties were in benzoid form. The geometries of *trans* form minima in S_1_, T_1_ and T_2_ states were quite similar to *trans*-S_0_. For *trans*-S_0_, the vertical excitation energy (VEE) to the S_1_ state (3.53 eV) was quite close to *trans*-S_1_ (3.23 eV) and *trans*-T_1_ (3.35 eV), and considering their similar geometry, there are competing relaxation pathways in S_1_ and T_1_ states.

In *cis*-S_0_, the DB moiety was very flat with C7=C8 of 1.326 Å and the benzene rings in NP and DAP twisted ~45° to eliminate the repulsion between them. Similar to *trans*-T_2_, the C7-C8 in *cis*-T_2_ was a double bond and as a fact of the localized electronic excitation in NP, the twisting of the DAP and NP moieties were nonsymmetrical, with α_Ph-A_ = 158.66° and α_Ph-N_ = 127.58°, respectively. In S_0_, S_1_, T_1_ and T_2_ states, we also obtained the DHP form minima, which were the ring-closing product of *cis* conformers on the respective state. For these DHP isomers, the π-conjugation system broke down due to the redundant H15 and H26 atoms, which resulted in a distorted phenanthrene plane. The phenanthrene ring in DHP-S_0_ occupied a polyene configuration with alternating single and double bonds. The aromaticity of DHP-S_1_, DHP-T_1_ and DHP-T_2_ was stronger, and the C-C bond lengths in the phenanthrene ring were more averaged in comparison with DHP-S_0_.

For the S_1_ and T_1_ states, the DANS molecule could twist towards the evidently weakened C7–C8 bond and thus the *twist*-S_1_ and *twist*-T_1_ achieved stable minima with similar *α*_C=C_. Moreover, the DMA moiety was planar in *twist*-S_1_, but pyramidal in *twist*-T_1_. In *twist*-S_1_, there were independent conjugated systems within quinoid DAP and NP, respectively, leading to the planar DMA moiety. However, in *twist*-T_1_, both NP and DAP were in typically benzoid form, and the DMA was pyramidalized. These midway minima along the *α*_C=C_ torsional path overlapped with the twisted form CIs and ISCs, and played important roles in nonadiabatic relaxation dynamics. The TS-S_0_ lay in the middle of the *α*_C=C_ torsional path with *α*_C=C_ = −92.20°, α_Ph-A_ = −177.51° and α_Ph-N_ = −177.89°, respectively. Moreover, the two methyl groups in DMA bent in the same direction without twisting around C12–N24. The ground state *cis* ↔ *trans* isomerization process could be divided into two parts, from *trans*-S_0_ to TS-S_0_, basically evolving along *α*_C=C_ torsion with slight pyramidalization of DMA, and from TS-S_0_ to *cis*-S_0_, there were synchronous twistings towards *α*_C=C_, *α*_Ph-A_ and *α*_Ph-N_.

### 3.2. Conical Intersections and Intersystem Crossings

The investigation on CIs and ISCs within low-lying electronic states of DANS is of crucial importance in unveiling the nature of nonadiabatic decay accompanied with *trans* ↔ *cis* isomerization and ring-closing reactions [67,68]. In previous studies, crossing regions were avoided and singlet–triplet quasi-degenerations were disclosed in one-dimensional PECs towards *α*_C=C_ [37]; however, the geometries and properties for possible CIs and ISCs involved in the photoisomerization and ring-closing of DANS are still unknown. In this work, we observed five S_1_/S_0_ and three T_2_/T_1_ CIs and three S_1_/T_2_, one S_1_/T_1_ and one S_0_/T_1_ ISCs at the SA6-CASSCF(18,12)/6-31G* level and the optimized geometries are depicted in Figure 1.

Four S_1_/S_0_ CIs were distributed along the S_1_
*trans* ↔ *cis* isomerization pathway with *α*_C=C_ of −31.98°, −109.14°, −96.74° and −154.63°, and labelled as CI-S_1_/S_0_-*cis*, CI-S_1_/S_0_-*twist-c*, CI-S_1_/S_0_-*twist-t* and CI-S_1_/S_0_-*trans*, respectively. In CI-S_1_/S_0_-*cis*, the C7–C8 bond became polarized and increased to 1.401 Å. The C7 atom was still in sp^2^ hybridization, and both C7 and H20 atoms stayed coplanar with an NP moiety. In contrast, the C8 converted to sp^3^ hybridization with H21 distortion to the rear side of C7–C8, similar as a hydrogen migration TS. Moreover, the DAP moiety also twisted towards C7–C8 with a *α*_Ph-A_ decrease to 129.04°. The DB moiety in CI-S_1_/S_0_-*trans* was similar to that in CI-S_1_/S_0_-*cis*, except that the pyramidalization of C8 atom was in the opposite direction. Accompanied with the DAP twisting towards the DANS molecule plane for 42.64°, the DMA turned perpendicular to the neighboring benzene plane. As a fact of the highly distorted geometry, the potential energies of CI-S_1_/S_0_-*cis* and CI-S_1_/S_0_-*trans* were ~1.00 eV higher than S_1_-S_0_ VEE of the respective ground state minima. Hence, these two CIs were not energetically accessible on gas phase dynamics upon S_1_ excitation. The highly distorted CI-S_1_/S_0_-*twist-c* and CI-S_1_/S_0_-*twist-t* were ~0.43 eV lower and ~0.45 eV higher than *trans*-S_1_-FC, respectively, but both were lower than *cis*-S_1_-FC with ~1.14 eV and ~0.25 eV. The polarization of C7 and C8 atoms also existed in CI-S_1_/S_0_-*twist-c* or CI-S_1_/S_0_-*twist-t*; however, pyramidalization took place at the C7 atom in the opposite direction and exhibited –65.98° and –132.53° of dihedral angle H20–C7–C8–H21, respectively. In both of them, the DAP moiety stayed coplanar with C7–C8, while the NP moieties twisted 22.75° and –30.42° towards C7–C8, respectively.

In vicinity of *trans*-S_1_, we optimized ISC-S_1_/T_2_-*trans*, which was only ~0.05 eV and ~0.35 eV higher than *trans*-S_1_-FC and *trans*-S_1_, respectively, and their representative internal coordinates were very similar. Moreover, the SOC at ISC-S_1_/T_2_-*trans* was 41.6 cm^−1^ and thus was an efficient relaxation channel. With *α*_Ph-A_ and *α*_Ph-N_ twisting for 18.01° and 19.41° and pyralmidalization of DMA, the CI-T_2_/T_1_-*trans* was observed at ~0.20 eV lower than ISC-S_1_/T_2_-*trans* and could serve as the intermediate for the S_1_ → T_1_ decay process. Twisting of the NP moiety alone created CI-T_2_/T_1_-*tict*, in which both NP and DAP moieties were planar and perpendicular to each other. However, the energy of CI-T_2_/T_1_-*tict* was ~1.03 eV higher than *trans*-S_1_, therefore, which was hardly achieved by *trans*-DANS upon S_1_ excitation.

Along the DB twisting coordinate started from the *cis*-S_1_-FC region, we optimized the CI-S_1_/S_0_-DHP, ISC-S_1_/T_1_-*cis* and ISC-S_1_/T_2_-*cis* with *α*_C=C_ values of −18.97°, −36.99° and −49.66°, respectively. Additionally, the SOC at ISC-S_1_/T_1_-*cis* and ISC-S_1_/T_2_-*cis* were both less than 1.0 cm^−1^, respectively. The peculiar CI-S_1_/S_0_-DHP and ISC-S_1_/T_1_-*cis* lay in the ring-closing pathway that led to the formation of DHP on S_0_ and T_1_ states, respectively. Similar reaction pathways have been reported for tetraphenylethylene [69]. The potential energies of CI-S_1_/S_0_-DHP and ISC-S_1_/T_1_-*cis* were ~1.87 and ~1.10 eV lower than *cis*-S_1_-FC, respectively. Within CI-S_1_/S_0_-DHP (ISC-S_1_/T_1_-*cis*), both of the two benzene rings twisted clockwise to approach each other with C1–C14 distance at only 1.984 (2.174) Å. Simultaneously, the C1 and C14 atoms showed evident an pyramidal structure with the C1–H15 and C14–H26 bonds bending out of the respective benzene ring plane for ~40 (~29)°. The ISC-S_1_/T_2_-*cis* lay at ~0.77 eV lower than *cis*-S_1_-FC with both NP and DAP moieties in planar form. The CI-T_2_/T_1_-*cis* lay in the vicinity of the *cis*-S_1_-FC region with *α*_C=C_ of −1.1°, in which the planar NP moiety stayed perpendicular to the coplanar DB and DAP moieties and the energy was ~0.32 eV lower than *cis*-S_1_-FC. The steep *α*_C=C_ twisting pathway with the reverse motion from low energy ISC-S_1_/T_2_-*cis* to high energy CI-T_2_/T_1_-*cis* along the reverse direction was unlikely to take place, and thus ISC-S_1_/T_2_-*cis* and CI-T_2_/T_1_-*cis* were not suitable partners to accomplish the S_1_ → T_1_ relay decay process within cisoid conformation. The alternative decay channel for T_2_ DANS after passing ISC-S_1_/T_2_-*cis* involved ISC-S_1_/T_2_-*twist* that lay in the middle of the *α*_C=C_ twisting pathway. However, the energy was ~0.72 eV higher than *cis*-S_1_-FC, and the SOC had an extremely weak strength of 0.2 cm^−1^; therefore, the ISC-S_1_/T_2_-*twist* was located beyond the accessible region of *cis*-DANS upon S_1_ excitation. Moreover, the high energy ISC-S_1_/T_2_-*twist* prevented further forward *α*_C=C_ twisting on the T_2_ surface and thus excluded the possibility of decay via transoid form CI-T_2_/T_1_-*tict* or CI-T_2_/T_1_-*trans*. For molecules in the T_1_ state, the decay channel to the S_0_ state passed through ISC-S_0_/T_1_-*twist*, which overlapped with the *trans*-T_1_ potential well and lay above the TS-S_0_.

### 3.3. The Relaxation Pathways for DANS Upon S_1_ Excitation

It is well accepted that the interplay of minima, CIs and ISCs determines the fate of excited-state molecule; therefore, the LIIC curves that connect those critical geometries present the intuitive view for the possible relaxation pathways. The MS-NEVPT2 calculated LIIC curves for CI-S_1_/S_0_-*twist-c*, CI-S_1_/S_0_-DHP and ISC-S_1_/T_1_-*cis* with the corresponding reactant and product are depicted in Figure 2a, Figure 2b and Figure 2c, respectively. The LIIC for multistep triplet relaxation pathway that involves ISC-S_1_/T_2_-*trans*, CI-T_2_/T_1_-*trans* and ISC-S_0_/T_1_-*twist* is presented in Figure 2d. Based on these LIIC curves, the qualitative analysis on the photoisomerization mechanisms of *trans*- and *cis*-DANS upon S_1_ excitation is presented.

After photoexcitation to *trans*-S_1_-FC, in-plane geometrical and electronical rearrangement took place and the molecule converted to the quinoid form. The subsequent relaxation may have been proceeding via intramolecular vibrational energy redistribution within *trans*-S_1_ potential well to populate enough energy to out-of-plane torsion modes and start the relaxation along *α*_C=C_ twisting coordinate. Another choice was stepwise decay to the T_1_ state via ISC-S_1_/T_2_-*trans* that overlapped with the *trans*-S_1_ potential well region and CI-T_2_/T_1_-*trans* and was followed by DB torsional relaxation in T_1_ state. A similar triplet relaxation channel was suggested by previous spectroscopy studies [13,14,18]. The relaxation started from *cis*-S_1_-FC was faster than those from *trans*-S_1_-FC because of disappearance of bound potential well around *cis*-S_1_-FC and the much steeper PES towards crossing regions. The simultaneous twisting of DAP and NP in the same or opposite direction gave rise to the competing DHP formation and *α*_C=C_ twisting relaxation pathways. Along ring closing, there was bifurcation into singlet or triplet pathway via CI-S_1_/S_0_-DHP and ISC-S_1_/T_1_-*cis*, respectively. Additionally, the *α*_C=C_ twisting pathway via ISC-S_1_/T_2_-*cis* was excluded because all the possible consecutive decays, via CI-T_2_/T_1_-*cis*, CI-T_2_/T_1_-*tict*, CI-T_2_/T_1_-*trans* or ISC-S_1_/T_2_-twist, were not favored under S_1_ excitation. DANS trapped in *twist*-S_1_ or *twist*-T_1_ potential well could return to the ground state and yield to either *cis*- or *trans*-S_0_ via CI-S_1_/S_0_-*twist-c* and ISC-S_0_/T_1_-*twist*, respectively.

The detailed information for relaxation dynamics is given as follows. Except for CI-S_1_/S_0_-*twist-c*, the other three CIs along S_1_
*α*_C=C_ twisting pathway of DANS, CI-S_1_/S_0_-*cis*, CI-S_1_/S_0_-*twist-t* and CI-S_1_/S_0_-*trans* lay beyond the capability of S_0_ → S_1_ vertical excitation of the adjacent *cis*-S_0_ or *trans*-S_0_ and thus prevented nonadibatic decay channels on gas phase. As shown in Figure 2a, although the energy of CI-S_1_/S_0_-*twist-c* was lower than *trans*-S_1_-FC, there was a considerable barrier between them, and such a topology of PEC agreed with previous DFT results [37]. With consideration of the constraint of planar conformation and more than 1.03 eV of barrier in the out-of-plane *α*_C=C_ twisting pathway, the singlet pathway for *trans*-S_1_-FC played a minor role. For the counterpart *cis*-S_1_-FC, there was a wide low barrier for accessing the CI-S_1_/S_0_-*twist-c* region. Nevertheless, due to the high initial potential energy, the *cis*-S_1_-FC → CI-S_1_/S_0_-*twist-c* relaxation could be achieved efficiently and serve as dominant isomerization pathway as in parent stilbene [70,71,72,73]. Additionally, the potential energy of CI-S_1_/S_0_-*twist-c* was ~0.65 eV higher than *twist*-S_1_, and the repeated oscillation toward the deep well before successful decay may result in evidently extended S_1_ lifetime. It should be noted that the T_1_ state stayed close to the S_1_ state at CI-S_1_/S_0_-*twist-c*, indicating that the competition between singlet and triplet pathways existed not only in the FC region. Another feasible singlet relaxation pathway for *cis*-S_1_-FC is given in Figure 2b, which corresponded to the ring-closing process on the S_1_ surface via CI-S_1_/S_0_-DHP. As a fact of the monotonous decreasing of PEC towards the crossing region, this decay channel was efficient, and after successful hopping to S_0_, the DANS molecule could persist the ring-closing process or back to *cis*-S_0_ as the CI-S_1_/S_0_-DHP was basically a barrier on S_0_ state.

The triplet relaxation pathways for *cis*-S_1_-FC and *trans*-S_1_-FC were rather different. At *trans*-S_1_-FC, it was a T_2_ mediated S_1_ → T_1_ process with relaxation along twisting coordinate on T_1_ surface, followed by the decay to ground state via ISC-S_0_/T_1_-*twist*. However, in the vicinity of the *cis*-S_1_-FC region, the only conformation and energy allowed triplet decay channel was the DHP formation via ISC-S_1_/T_1_-*cis*. As show in Figure 2c, the S_1_ PEC from *cis*-S_1_-FC to ISC-S_1_/T_1_-*cis* decreased mildly and thus favored a relaxation pathway with NP and DAP twisting in opposite directions. After a decay to the T_1_ state, the widespread degeneration between T_1_ and S_0_ states existed along the ring-closing procedure, which offered a possible T_1_ → S_0_ decay channel, as has been revealed in *o*-nitrophenol [74]. Moreover, this triplet pathway was still limited by insufficient SOC strength. In case the large amplitude *α*_C=C_ twisting on T_2_ surface was feasible, the CI-T_2_/T_1_-*trans* and CI-T_2_/T_1_-*tict* were both energetically accessible after the *cis*-S_1_-FC → ISC-S_1_/T_2_-*cis* process; unfortunately, as evidenced in Figure 2a, the twisting conformation on T2 surface was a barrier between cis- and twist-regions. The stepwise triplet isomerization pathway, trans-S1-FC ↔ ISC-S1/T2-trans ↔ CI-T2/T1-trans ↔ ISC-S0/T1-twist ↔ cis-S1-FC is presented in Figure 2d. Starting from trans-S1-FC, the potential energy decreased mildly until the arrival of CI-T2/T1-trans and then became steep towards ISC-S0/T1-twist. As the internal conversion was usually more efficient than ISC, the ISC-S_0_/T_1_-*twist* (1.9 cm^−1^) with much smaller SOC value in comparison with ISC-S_1_/T_2_-*trans* (41.6 cm^−1^) was the rate-controlling step. Although the SOC at ISC-S_0_/T_1_-*twist* was weak, the decay event could still take place in case that there were enough times of repeated attempts and suitable coupling between crossing states.

## 4. Conclusions

We investigated the photorelaxation mechanisms of DANS upon S_1_ excitation by constructing the interwoven conical intersection and intersystem crossing networks at the SA6-CASSCF//MS-NEVPT2/6-31G* level. A schematic plot for the possible relaxation routes is displayed in Figure 3 and competitions within them are presented as follows.

After excitation to the *trans*-S_1_-FC, with in-plane electronic and geometry rearrangement, the DANS quickly convert to quinoid conformation. Subsequently, the relaxation mainly takes place along the triplet pathway ISC-S_1_/T_2_-*trans* → CI-T_2_/T_1_-*trans* → ISC-S_0_/T_1_-*twist* → *trans*- or *cis*-S_0_. Another competitive triplet pathway via CI-T_2_/T_1_-*tict* is unlikely taking place for its high energy. Nevertheless, the singlet pathway, *trans*-S_1_-FC → CI-S_1_/S_0_-*twist-c* → *trans*- or *cis*-S_0_, contribute a little as it is hindered by a rather high barrier on the S_1_ surface between *trans*-S_1_-FC and CI-S_1_/S_0_-*twist-c*. The other conical regions on the S_1_ surface, CI-S_1_/S_0_-*trans*, CI-S_1_/S_0_-*twist-t* and CI-S_1_/S_0_-*cis*, stay away from the accessible region of DANS started from *trans*-S_1_-FC. The mechanisms for *trans*-DANS presented in this work are consistent with the experimentally observed ~0.1 singlet-triplet branching ratio in nonpolar solvents [68]. For DANS excited to *cis*-S_1_-FC, two singlet pathways, *cis*-S_1_-FC → CI-S_1_/S_0_-*twist-c* → *trans*- or *cis*-S_0_ and *cis*-S_1_-FC → CI-S_1_/S_0_-DHP → DHP-S_0_, decay along the steeply descending S_1_ surface and thus are the dominate pathways. The fate of the *cis*-form DANS on the S_1_ surface is determined by the coupling of the initial twisting direction of DN and DAP. A similar ring-closing pathway observed on the T_1_ surface is *cis*-S_1_-FC → ISC-S_1_/T_1_-*cis* → DHP-T_1_ → DHP-S_0_, which is less important compared with the singlet counterpart due to the weak SOC strength. The other two possible triplet pathways mediated by ISC-S_1_/T_2_-*cis* are *cis*-S_1_-FC → ISC-S_1_/T_2_-*cis* → CI-T_2_/T_1_-*cis* → ISC-S_0_/T_1_-*twist* → *trans*- or *cis*-S_0_ and *cis*-S_1_-FC → ISC-S_1_/T_2_-*cis* → ISC-S_1_/T_2_-*twist* → CI-S_1_/S_0_-*twist-c* → *trans*- or *cis*-S_0_, which are basically impossible due to geometry and energy incompatibility. To quantitatively reveal the lifetimes of involved excited state intermediates, quantum yields for photoisomerization, interplay and branch ratios between different relaxation channels, we are continuing with full-dimensional nonadiabatic trajectory surface hopping molecular dynamics simulations with potential energies and gradients calculated by SA6-CASSCF, and the results will be presented in our forthcoming work.

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
