# Peer review of "Photorelaxation Pathways of 4-(N,N-Dimethylamino)-4′-nitrostilbene Upon S1 Excitation Revealed by Conical Intersection and Intersystem Crossing Networks"

_molecules, 2020, doi:10.3390/molecules25092230_

Round 1
Reviewer 1 Report
This manuscript presents a computational study of the photophysics and photochemistry of the title compound (DANS), a molecule of interest as a model photoswitch that has been broadly studied experimentally and theoretically. The authors attempt to do a full characterization of the PES using MS-NEVPT2 on optimized CASSCF geometries. Overall, the main message of the manuscript is clear and appears to be consistent with the experimental results. Therefore, the work has broad interest and is potentially suitable for Molecules. However, the active space used in the CASSCF calculations is hard to justify and definitely improvable (see detailed comments). Since the active space choice is always a key aspect in this type of calculations, this point of the manuscript has to be improved. A reasonable compromise would be to recalculate the energies of key critical points with a better active space to confirm that the results are not flawed by this issue. Therefore, I recommend that the authors carry out this major revision and resubmit the manuscript for further review.
Detailed comments:
(1) The choice of an (18,12) active space is quite puzzling. First, an active space made of 9 occupied and 3 empty orbitals is clearly unbalanced. It is also surprising that the authors include 4 sigma orbitals centered on the nitrobenzene moiety (HOMO-5 to HOMO-8 in Figure S1) which probably have occupations very close to 2.00 and will not be contributing to the multireference character of the wave function. In other words, this active space is effectively a (10,8) active space because the sigma orbitals will not play any role. Therefore the methodology cannot be considered state-of-the-art.
The authors have to demonstrate that the conclusions obtained using this active space are still valid, and to do this I suggest that they calculate the energies of key critical points with a more carefully chosen (12,12) active space. I recommend to consider the following structures: the cis and trans S0 minima, CI-S1/S0-twist-c and some point along the reaction coordinate in Figure 2a, ISC-S1/T1-cis, ISC-S1/T2-trans and ISC-S0/T1-twist. These calculations should be well feasible from the point of view of the required computational resources.
(2) The authors postulate (Figure 2b) that the formation of the DHP product involves the ISC-S1/T1-cis crossing. However the energy profile suggests that this can occur directly through an S1/S0-CI (similar to what happens in tetraphenylethylene, see https://doi.org/10.1039/C5CP04546K). I recommend the authors to optimize this CI and add this possibility to their mechanistic description.
Author Response
Referee 1:
(1) The choice of an (18,12) active space is quite puzzling. First, an active space made of 9 occupied and 3 empty orbitals is clearly unbalanced. It is also surprising that the authors include 4 sigma orbitals centered on the nitrobenzene moiety (HOMO-5 to HOMO-8 in Figure S1) which probably have occupations very close to 2.00 and will not be contributing to the multireference character of the wave function. In other words, this active space is effectively a (10,8) active space because the sigma orbitals will not play any role. Therefore the methodology cannot be considered state-of-the-art.
The authors have to demonstrate that the conclusions obtained using this active space are still valid, and to do this I suggest that they calculate the energies of key critical points with a more carefully chosen (12,12) active space. I recommend to consider the following structures: the cis and trans S0 minima, CI-S1/S0-twist-c and some point along the reaction coordinate in Figure 2a, ISC-S1/T1-cis, ISC-S1/T2-trans and ISC-S0/T1-twist. These calculations should be well feasible from the point of view of the required computational resources.
Reply: Thanks to referee for this comment. We have done comparison calculations with different active spaces for the crucial points of trans-S0 and cis-S0, including CAS(12,12) suggested by referee. The active molecular orbitals of them for trans-S0 and cis-S0 are presented in Figure S1 on the supporting information. It is evident that the corresponding orbitals of CAS(18,12) for trans-S0 are in agreement with CAS(10,8) but different from CAS(10,10) and CAS(12,12). The latter orbitals are more balance in the treatment of π and π* orbitals, however CAS(18,12) for trans-S0 have four σ orbitals and electrons on the electron withdrawing group side are more than those on the side of the electron donating group. On the other hand, CAS(18,12) for cis-S0 has only one σ orbitals and the remaining π and π* orbitals are delocalized with electron distributions on both sides. It tends to attribute the different electron distributions of trans-S0 and cis-S0 to the molecular geometry. The planar geometry of trans-S0 leads to strong electronic transfer for active orbitals of CAS(18,12) and CAS(10,8), while the nonplanar geometry of cis-S0 hinders this transfer and maintains more delocalized distribution. The vertical excitation energies from trans-S0 to the first excited state S1 calculated by MS-NEVPT2 with reference wavefunctions from CAS(10,10) and CAS(12,12) are 4.16 eV and 4.95 eV, respectively, which are remarkably larger than 3.62 eV from CAS(10,8) and 3.53 eV from CAS(18,12) and then 2.97 eV from experiments in cyclohexane [See ref. 14]. Since the energy of CAS(18,12) is the closest one to experimental observation and the initial excitation energy plays an important role in the photochemical reaction, this active space is feasible for computations in this work. This discussion for the choice of active space has been added to this paper.
(2) The authors postulate (Figure 2b) that the formation of the DHP product involves the ISC-S1/T1-cis crossing. However the energy profile suggests that this can occur directly through an S1/S0-CI (similar to what happens in tetraphenylethylene, see https://doi.org/10.1039/C5CP04546K). I recommend the authors to optimize this CI and add this possibility to their mechanistic description.
Reply: Thanks for the suggestions, we have optimized the CI-S1/S0-DHP geometry which is quite similar to ISC-S1/T1-cis. This CI is energetically favored for cis-S1-FC and should play an important role in photorelaxation and competing with the other singlet and triplet reaction pathways. The discussions for this decay channel was added to the manuscript. Moreover, the suggested reference paper was also cited.
Reviewer 2 Report
The manuscript deals with the excited relaxation of an organic molecule with a central C=C double bound, two phenyl rings and N(CH3)3 or NO2 side groups. The topic is very interesting and the calculations are mostly correctly performed. As it is always very difficult to organize the large amount of data generated in these studies, it is nice to see that the authors have made an attempt to summarise the most important results in a singlet graph at the end of the article. The article can be accepted for publication but requires one important clarification, which I describe after an initial list of minor points that should also be considered.
-In line 336 it is stated that after excitation, the trans-FC-S1 converts quickly to quinoid trans-S1. I do not fully understand what is meant here. Neither in figure 2a nor 2c, a fast relaxation path can be observed for the FC-S1 state. It could be that the authors are describing a change in the electronic character, rather than a geometrical relaxation?
-Is the S1 the state of maximum transition probability, that is the bright state that will be have the largest population upon excitation?
-The authors might want to comment the fact that the ISC and CI geometries are obtained with a different method (CASSCF) than the method that is used to calculate the PEC connecting different critical points by LIIC. Note that the energy at ISC and CI are not strictly degenerate. In most cases the difference is small but there are a few exceptions. For example in the case of the CI-T2/T1-trans, T1 and T2 are separated by 0.7 eV.
-Without having to consult the SI, it would be nice to have a short note on the importance of NEVPT2 for the interpretation of the relaxation mechanism. Just a short sentence stating that PECs calculated with CASSCF are substantially different (or not...)
-The SI could be completed with a ball-and-stick representation of the optimized minima, especially the DHP forms are so different from the others that a picture may be help to easier understand the text.
-The headers of the last two columns of Table 1 do not correspond to the labels in scheme 1
These minor points bring me to my main point, namely the choice of the active space and the shape of the active orbitals depicted in the SI. There are two issues with the active space. In the first place, the authors state that the 12 orbitals in the CAS correspond to pi orbitals, but the first four orbitals in fig S1 are sigma orbitals. For HOMO-5, -6 and -7 this is very easily seen from the figure and the shape of HOMO-8 also strongly suggest sigma character. Secondly, the three orbitals labeled with LUMO, LUMO+1 and LUMO+2 are all three localized on the right of the molecule. This will almost certainly introduce an unbalance in the treatment of the dynamic correlation by NEVPT2. The right phenyl ring is much better described with these active orbitals than the left ring. In molecules with conjugated pi systems, it is common practice to have an active space with as many active orbitals as active electrons. This means that for each pi orbital, doubly occupied in the Hartree-Fock determinant, there is another pi* orbital in the active space. Examples of these correlating pairs of active orbitals are HOMO-1/LUMO+1 and HOMO-2/LUMO+2. The authors should either give very strong, convincing arguments why this 'strange' active space is justified in this case, or they should repeat (some of) the calculation with a better balanced active space such as CAS(10,10) or CAS(12,12) or larger if required for a good description of all the excited states.
Author Response
Referee 2:
-In line 336 it is stated that after excitation, the trans-FC-S1 converts quickly to quinoid trans-S1. I do not fully understand what is meant here. Neither in figure 2a nor 2c, a fast relaxation path can be observed for the FC-S1 state. It could be that the authors are describing a change in the electronic character, rather than a geometrical relaxation?
Reply: The descriptions for related paragraphs were revised. We intend to illustrate the initial electronic and geometric rearrangement within molecule after excited to S1 state.
-Is the S1 the state of maximum transition probability, that is the bright state that will be have the largest population upon excitation?
Reply: The S1 state has the largest transition probability in the low-lying state as denoted in the ref. 14 (J. Phys. Chem. A 1989, 93, 7144–7152).
-The authors might want to comment the fact that the ISC and CI geometries are obtained with a different method (CASSCF) than the method that is used to calculate the PEC connecting different critical points by LIIC. Note that the energy at ISC and CI are not strictly degenerate. In most cases the difference is small but there are a few exceptions. For example in the case of the CI-T2/T1-trans, T1 and T2 are separated by 0.7 eV.
Reply: The geometry of CASSCF optimized crossing may be different with the real crossing at NEVPT2, but may not be far away from each other. It can be found in Figure 2 that the LIIC curves have crossing points at MS-NEVPT2 level, which is close to the crossing points for ISC and CI of CASSCF results. The energy differences for crossing points of NEVPT2 curves are all less than 0.3 eV in Figure 2, which may be smaller than the energy gaps calculated by MS-NEVPT2 at the geometry of CASSCF crossing points, particularly 0.29 eV for NEVPT2 crossing point in Figure 2b and 0.64 eV for CASSCF geometry of CI-T2/T1-trans in Table 4.
-Without having to consult the SI, it would be nice to have a short note on the importance of NEVPT2 for the interpretation of the relaxation mechanism. Just a short sentence stating that PECs calculated with CASSCF are substantially different (or not...)
Reply: The explanations for why the NEVPT2 utilized to analyze the reaction pathways is that the CASSCF method evidently overestimates the vertical excitation energies, which were added to manuscript from line 122 to 128.
-The SI could be completed with a ball-and-stick representation of the optimized minima, especially the DHP forms are so different from the others that a picture may be help to easier understand the text.
Reply: The geometries of optimized minima were displayed in Figure S3.
-The headers of the last two columns of Table 1 do not correspond to the labels in scheme 1
Reply: The headers in Table 1 was revised according to Scheme 1.
These minor points bring me to my main point, namely the choice of the active space and the shape of the active orbitals depicted in the SI. There are two issues with the active space. In the first place, the authors state that the 12 orbitals in the CAS correspond to pi orbitals, but the first four orbitals in fig S1 are sigma orbitals. For HOMO-5, -6 and -7 this is very easily seen from the figure and the shape of HOMO-8 also strongly suggest sigma character. Secondly, the three orbitals labeled with LUMO, LUMO+1 and LUMO+2 are all three localized on the right of the molecule. This will almost certainly introduce an unbalance in the treatment of the dynamic correlation by NEVPT2. The right phenyl ring is much better described with these active orbitals than the left ring. In molecules with conjugated pi systems, it is common practice to have an active space with as many active orbitals as active electrons. This means that for each pi orbital, doubly occupied in the Hartree-Fock determinant, there is another pi* orbital in the active space. Examples of these correlating pairs of active orbitals are HOMO-1/LUMO+1 and HOMO-2/LUMO+2. The authors should either give very strong, convincing arguments why this 'strange' active space is justified in this case, or they should repeat (some of) the calculation with a better balanced active space such as CAS(10,10) or CAS(12,12) or larger if required for a good description of all the excited states.
Reply: Thanks to referee for this comment and suggestion. Some common complete active spaces, such as CAS(10,8), and suggested CAS(10,10) and CAS(12,12), have been chosen to compare with CAS(18,12) in this work. As shown in Figure S1 on the supporting information for active orbitals, it is evident that the corresponding orbitals of CAS(18,12) for trans-S0 are in agreement with CAS(10,8) but different from CAS(10,10) and CAS(12,12). The latter orbitals are more balance in the treatment of π and π* orbitals, however CAS(18,12) for trans-S0 have four σ orbitals and electrons on the electron withdrawing group side are more than those on the side of the electron donating group. On the other hand, CAS(18,12) for cis-S0 has only one σ orbitals and the remaining π and π* orbitals are delocalized with electron distributions on both sides. It tends to attribute the different electron distributions of trans-S0 and cis-S0 to the molecular geometry. The planar geometry of trans-S0 leads to strong electronic transfer as shown in Figure S1 for active orbitals of CAS(18,12) and CAS(10,8), while the nonplanar geometry of cis-S0 hinders this transfer and maintains more delocalized distribution. The vertical excitation energies from trans-S0 to the first excited state S1 calculated by MS-NEVPT2 with reference wavefunctions from CAS(10,10) and CAS(12,12) are 4.16 eV and 4.95 eV, respectively, which are remarkably larger than 3.62 eV from CAS(10,8) and 3.53 eV from CAS(18,12) and then 2.97 eV from experiments in cyclohexane [See ref. 14]. Since the initial excitation energy plays an important role in the photochemical reaction, the energy of CAS(18,12) is the closest one to experimental observation so that this active space is feasible for computations in this work. This discussion for the choice of active space has been added to this paper.
Round 2
Reviewer 1 Report
In response to my comments, the authors provide the MS-NEVPT2 vertical excitation energies calculated with several CASSCF reference active spaces. The results are spread over a range of 3.53 to 5.01 eV, which is worrying. I would expect much smaller variations of the order of some tenths of eV. This suggests that there may be some problems with the method that are being overlooked by the authors.
That being said, the mechanistic picture seems OK so that the manuscript can be considered suitable for publication from the point of view.
Besides, Table S1 in the SI is missing. The uploaded SI starts with Table S2.
Author Response
Reply: Thanks to referee for the suggestion. We have added Table S1 and S3 to display the occupation numbers of active orbitals for the stationary points by SA6-CASSCF(18,12) method, respectively.
Reviewer 2 Report
The revision of the manuscript has shed some light on the peculiar choice of active space. I still don't fee comfortable with the CAS(18,10), but at least there are some arguments in the paper that may be something like a justification. In fact, the argument given by the authors is that the NEVPT2 excitation energies calculated withe the CAS(18,12) active space give the best match with experiment. Using this as an argument is to some extent dangerous as the agreement may be the effect of cancellation of errors. In this case an unbalanced active space combined with a rather small basis one-electron basis set could possibly lead to good answers for the wrong reason. Even after the revision, in which the authors indeed performed some calculations with more balanced active spaces, there are still reasonable doubts about the role of the formally doubly occupied sigma orbitals in the CAS(18,10). The natural occupation numbers of these orbitals must be very close to two and in fact, as mentioned by the other reviewer, these orbitals do not actively play a role in the multiconfigurational reference wave function, reducing the CAS(18,10) effectively to a CAS(10,8). For some reason, these sigma orbitals turn into pi orbitals in the cis geometry, but still then there are probably very little determinants in the CASSCF wave function in which these low-lying pi orbitals are not doubly occupied. As a solution, and as a way to check the validity of the CAS for the reader, I would strongly suggest that the article (or supp info) is extended with two extra points: (i) a table in which the natural occupation numbers of the active orbitals at some key geometries are discussed. In case there are many orbitals with occupation number larger than 1.99 (which is probably so), it may indicate that the choice of active space is not optimal, even in the cis geometry where al but one active orbital is of pi character. (ii) a remark that the small one-electron basis set may influence the excitation energies.
With these two additions, the manuscript can be accepted for publication.
Author Response
Reply: Thanks to referee for the suggestions.
(1) We have added Table S1 and S3 in SI to display the occupation numbers of active orbitals for the stationary points by SA6-CASSCF(18,12) method, respectively. Additionally, the occupation numbers of active orbitals for trans-S0 and cis-S0 by different active spaces have also been added in Figure S1. It can be seen in Table S1 and S3 that electronic excitations lie in the region between HOMO-2 and LUMO+2 with occupation numbers between ~1.80 and ~0.20. It is interesting that the occupation numbers of different active orbitals in Figure S1 have also provided the same region for electronic excitations. The large active space CAS(12,12) has high-lying LUMO+4 and LUMO+5 with occupation numbers less than ~0.1, which has the trivial contribution to the main configuration of low-lying electronic states. Consequently, CAS(18,12) has contained necessary active orbitals for the description of electronic states involved in this work. Moreover, the influence on S0→S1 VEE by active spaces can be explained by the electronic configuration of S1 state. For both CAS(18,12) and CAS(10,8), the main configuration of S1 corresponded to the HOMO→LUMO excitation with the weight of ~0.70, however, which reduces to only 0.51 in CAS(10,10). Conversely, for CAS(12,12), the main configurations of S1 are HOMO→LUMO+2 (0.24), HOMO-3→LUMO+1 (0.18) and HOMO-2→LUMO+2 (0.12), respectively. The small and nearly zero weights of HOMO→LUMO in CAS(10,10) and CAS(12,12) yield to high excitation energies of 4.16 eV and 4.95 eV, respectively. This discussion for the choice of active space has been added to Section 2 in this paper highlighted by blue color.
(2) Besides 6-31G*, we have also performed comparative calculations with larger basis sets of 6-311G* and 6-311+G* for S0→S1 VEEs at trans-S0 by MS-NEVPT2 with reference wavefunctions from SA6-CASSCF(18,12). The S0→S1 VEEs by 6-31G*, 6-311G* and 6-311+G* are 3.53 eV, 3.37 eV and 3.14 eV, respectively, which are gradually approaching the experimental value of 2.97 eV with enlarging basis sets. Although the small basis set 6-31G* leads to 0.39 eV overestimation from that of 6-311+G*, which agrees with the convergence trend to experimental value by extending basis set. With the consideration of computations time cost by employing those two larger basis sets, 6-31G* is available basis set in this work. This discussion for the choice of basis set has been added to Section 2 in this paper highlighted by blue color.